# Health facility availability and readiness for family planning and maternity and neonatal care services in Nepal: Analysis of cross-sectional survey data

**Pramila Rai** [ID], **Ilana N. Ackerman** [ID][☉], **Denise A. O'Connor**[☉], **Alexandra Gorelik** [ID], **Rachelle Buchbinder** [ID]*[☉]

School of Public Health and Preventive Medicine, Monash University, Melbourne, Victoria, Australia

☉ These authors contributed equally to this work.
* rachelle.buchbinder@monash.edu

**Data Availability Statement:** The data used in this study are owned by the Demographic and Health

## Abstract

### Objectives

To determine the availability and readiness of health facilities to provide family planning, antenatal care and basic emergency obstetric and newborn care in Nepal in 2021. Secondary objectives were to identify progress since 2015 and factors associated with readiness.

### Method

This is a secondary analysis of cross-sectional Nepal Health Facility Survey (NHFS) data collected in 2015 and 2021. The main outcome measures were availability and readiness of family planning, antenatal care, and basic emergency obstetric and newborn care services. Readiness indices were calculated using WHO-recommended service availability and readiness assessment (SARA) methods (score range 0 to 100%, with 100% indicating facilities are fully prepared to provide a specific service). We used independent t-tests to compare readiness indices in 2015 and 2021. Factors potentially associated with readiness (rurality setting, ecological region, managing authority, management meeting, quality assurance activities, and external supervision) were explored using multivariable linear regression.

### Results

There were 940 and 1565 eligible health facilities in the 2015 and 2021 surveys, respectively. Nearly all health facilities provided family planning (2015: n = 919 (97.8%); 2021: n = 1530 (97.8%)) and antenatal care services (2015: n = 920 (97.8%); 2021: n = 1538 (98.3%)) in both years, but only half provided delivery services (2015: n = 457 (48.6%); 2021: n = 804 (51.4%)). There were suboptimal improvements in readiness indices over time: (2015–21: family planning 68.0% to 70.9%, p<0.001, antenatal care 49.5% to 54.1%, p<0.001 and basic emergency obstetric and newborn care 56.7% to 58.0%, p = 0.115). The regression model comprising combined datasets of both NHFSs indicates facilities with regular management meetings and/or quality assurance activities had significantly greater readiness for

Surveys (DHS) Program and re-distribution of DHS data is not permitted by the DHS program. These data are available upon request and approval from the DHS Program. Data requests can be made through the link: https://dhsprogram.com/data/new-user-registration.cfm.

**Funding:** The authors received no specific funding for this work. Ms Rai is supported by a Monash University International Postgraduate Research Scholarship and Graduate Scholarship for her doctoral research. Dr Buchbinder is supported by an Australian NHMRC Investigator Fellowship (APP1194483). The funders had no role in study design, data collection and analysis, decision to publish, or preparation of the manuscript.

**Competing interests:** The authors have declared that no competing interests exist.

all three indices. Similarly, public facilities had greater readiness for family planning and basic emergency obstetric and newborn care while they had lower readiness for antenatal care.

## Conclusions

Readiness to deliver family planning, antenatal care and basic emergency obstetric and newborn care services in Nepal remains inadequate, with little improvement observed over six years.

## Introduction

About 15% of expected births worldwide result in life-threatening complications during the childbirth or the postpartum period [1]. Sometimes these complications develop rapidly and without warning, requiring immediate care [2–5]. According to the latest available data, maternal complications such as postpartum haemorrhage, puerperal sepsis, pre-eclampsia, eclampsia, obstructed or prolonged labour, and complications associated with unsafe abortion account for nearly 75% of maternal deaths globally [6, 7]. These issues are particularly relevant for lower and lower-middle income countries, such as Nepal, as 94% of the maternal deaths occur in these regions [8].

The highly acclaimed safe motherhood program [9, 10], which includes family planning, antenatal care, basic and comprehensive emergency maternity care, and postpartum care is not new to the Nepali health care system [11–13]. The safe motherhood program has been a central component of healthcare delivery in Nepal since the 1990s, contributing to significant improvements in maternal and neonatal health status over the last three decades [14]. In 2008, family planning alone was estimated to prevent 53% of maternal deaths in Nepal [15–17].

However, current health indicators such as the maternal mortality rate (239/1,000,000 live births), neonatal mortality rate (21/1,000 live births), and low contraceptive use (43% of reproductive aged women or their partners), indicate that improvements in existing health programs and their delivery are still needed [18, 19]. The persistent burden of maternal and neonatal mortality in Nepal (despite an increasing proportion of institutional births and antenatal care in recent years), also questions the quality of the services provided [20, 21]. Over 50% of maternal deaths in Nepal occur in health facilities or during the transition to health facilities, highlighting the need for service improvement [22].

Provision of high quality care for all women and newborn babies could prevent up to 54% of maternal deaths and 71% of neonatal deaths globally by 2025 [23]. It is also estimated that 30% of maternal deaths could be averted if the global need for contraception was met [16]. Unfortunately, several studies reporting cross-sectional data from the 2015 Nepal Health Facility Survey (NHFS) data found that Nepali health facilities are not yet well prepared to provide these essential maternal and newborn health services [24–26]. Based upon readiness indices where 100% indicates complete readiness, they reported that readiness to provide family planning services, antenatal care, and basic emergency obstetric and newborn care was only 68%, 49%, and 52%, respectively [24, 25].

In a study evaluating the technical quality of healthcare services in Nepal (using data from the 2015 NHFS), the average quality of care was only 0.55 (maximum score of 1.00) for both antenatal and perinatal care [27]. Private facilities were found to provide higher quality antenatal and perinatal services, compared to public facilities [27]. Other research has examined the

availability of equipment and medications for labour and neonatal care, and newborn care practices in Nepal and found substantial variation between hospital settings (including between public and private hospitals) [26]. These data provide a strong impetus for improvement and in response, the Ministry of Health and Population of Nepal rolled out Minimum Service Standard (MSS) tools designed to improve health facility preparedness for providing specific services [28].

We are not aware of any studies that have comprehensively examined whether preparedness for providing maternal and newborn services has improved since 2015. Initial reports using the 2021 NHFS data have focused on the readiness and availability of basic emergency obstetric and neonatal care only [29], or have provided only a high-level overview of family planning, antenatal, delivery and newborn care practices [30]. Changes in the provision of various family planning modalities from 2015 to 2021 have also been briefly described [30]. The present study aimed to assess changes over time in the readiness and availability of family planning, antenatal care, and delivery services, using data from the 2015 and 2021 Nepal Health Facility Surveys, and to identify factors associated with readiness.

## Methods

### Data source

The NHFS 2015 and 2021 data for this project were retrieved between May-September 2022 and followed the standard procedures for data use. Access to the data was granted to the first author (PR) by the DHS program for this specific project.

### Nepal Health Facility Survey (NHFS)

The NHFS survey was first conducted in 2015 and a second survey was conducted in 2021. Reports from both surveys have been published previously [30–32]. Each survey was conducted by New ERA [33], a local research firm, in partnership with the Nepal Ministry of Health. Technical assistance for survey implementation was provided by the Inner-City Fund International under the Demographic and Health Survey program with financial support from United States Agency for International Development and UK Department for International Development.

### Sample and sampling procedure

The sampling procedure and the data collection methods used in the Nepal Health Facility Survey have been reported in detail elsewhere [30–32]. Using a stratified random sampling design, the 2015 survey sampled a total of 1,000 health facilities from a master list of 4,719 formal-sector health facilities in Nepal. Of these, eight were duplicates and the final sample size was 992.The 2021 survey sampled 1,633 health facilities from a master list of 5,681 (excluding polyclinics). Of these, seven were duplicates, resulting in a final sample size of 1,626. We excluded standalone HIV testing and counselling sites and sites with incomplete survey data that resulted 940 facilities in 2015 and 1,565 facilities in 2021. The facility types were hospitals, primary health care centres, health posts, community health units, standalone HIV testing and counselling sites and urban health centres.

### Data collection methods

The NHFS utilised four types of identical survey instruments in both 2015 and 2021: 1) a facility inventory questionnaire; 2) a health provider questionnaire for individual health providers; 3) observation protocols for antenatal care, family planning, and services for sick children in

2015 and in 2021 labour and normal delivery were also included; and 4) exit interview questionnaires for antenatal care and family planning clients, women undergoing vaginal delivery, and carers of sick children whose consultations were observed by interviewers as part of the assessment. For this study, we analysed variables from the facility inventory questionnaire and the health provider questionnaire relating to family planning, antenatal care and basic emergency obstetric and newborn care.

## Measurement of variables

We examined two outcome measures: 1) availability of family planning, antenatal care and delivery services; and 2) service readiness (willingness or preparedness) of the facilities to provide these care as measured by the Family Planning Readiness Index for family planning services, the Antenatal Care Readiness Index for antenatal care service, and the Basic emergency obstetric and newborn care Readiness Index for delivery services including basic emergency obstetric and newborn care signal functions [34]. The seven signal functions of basic emergency obstetric and newborn care include parenteral administration of oxytocin, anticonvulsants and antibiotics, the performance of assisted deliveries, removal of retained products of conception, manual removal of retained placentae, and newborn resuscitation. These signal functions are vital interventions for the timely management of complications in mothers and newborns after childbirth. We described the signal functions provided by the health facilities in the last 3 months of the survey.

The readiness indices were derived based on WHO-recommended Service Availability and Readiness Assessment methodology by counting the physical presence of the essential items (called 'tracer items') required to deliver those specific services [24, 34]. The list of tracer items and the methods used are shown in the S1 Table. Operational definitions for service availability for family planning, antenatal care and basic emergency obstetric and neonatal care are presented in Text Box 1.

Each of the services comprises three specific domains with equal weight: 1) staff and guidelines, 2) equipment, and 3) medicines and commodities. The Antenatal care Readiness Index includes an additional equally weighted domain of 'diagnostics'. These domains are given an equal percentage weight of 33.33% when calculating indices for the family planning and basic emergency obstetric and newborn care readiness, and a 25% percentage weight when calculating the index for antenatal care. We calculated domain scores for each specific health service based on the presence of tracer items. Finally, mean specific readiness indices (range 0–100%) were calculated by taking the average of the included domain scores as percentages. The specific readiness index of 100% is considered fully prepared to provide a specific service, whereas 0 is the absence of all the required items for the service.

## Independent variables

For this study, we categorised health facilities according to ecological regions (i.e. mountain, hill, terai), provinces, settings (i.e. rural/urban), managing authority (public/private) and facility levels (i.e. peripheral public facilities with basic health facilities serving as the first contact point for patients, advanced public health facilities providing comprehensive health services and acting as referral centres for peripheral health facilities, and private hospitals managed by private authorities). For the 2015 dataset, we utilised global positioning system (GPS) location information to derive information on provinces and to stratify health facilities into rural if they were in a village municipality or urban if they were located in a municipality, sub-metropolitan or metropolitan city. These rural/urban data were provided in the 2021 dataset.

Text Box 1. Operational definitions of service-specific availability and readiness indices for family planning, antenatal care and delivery services

*BEmONC: Basic emergency obstetric and newborn care.

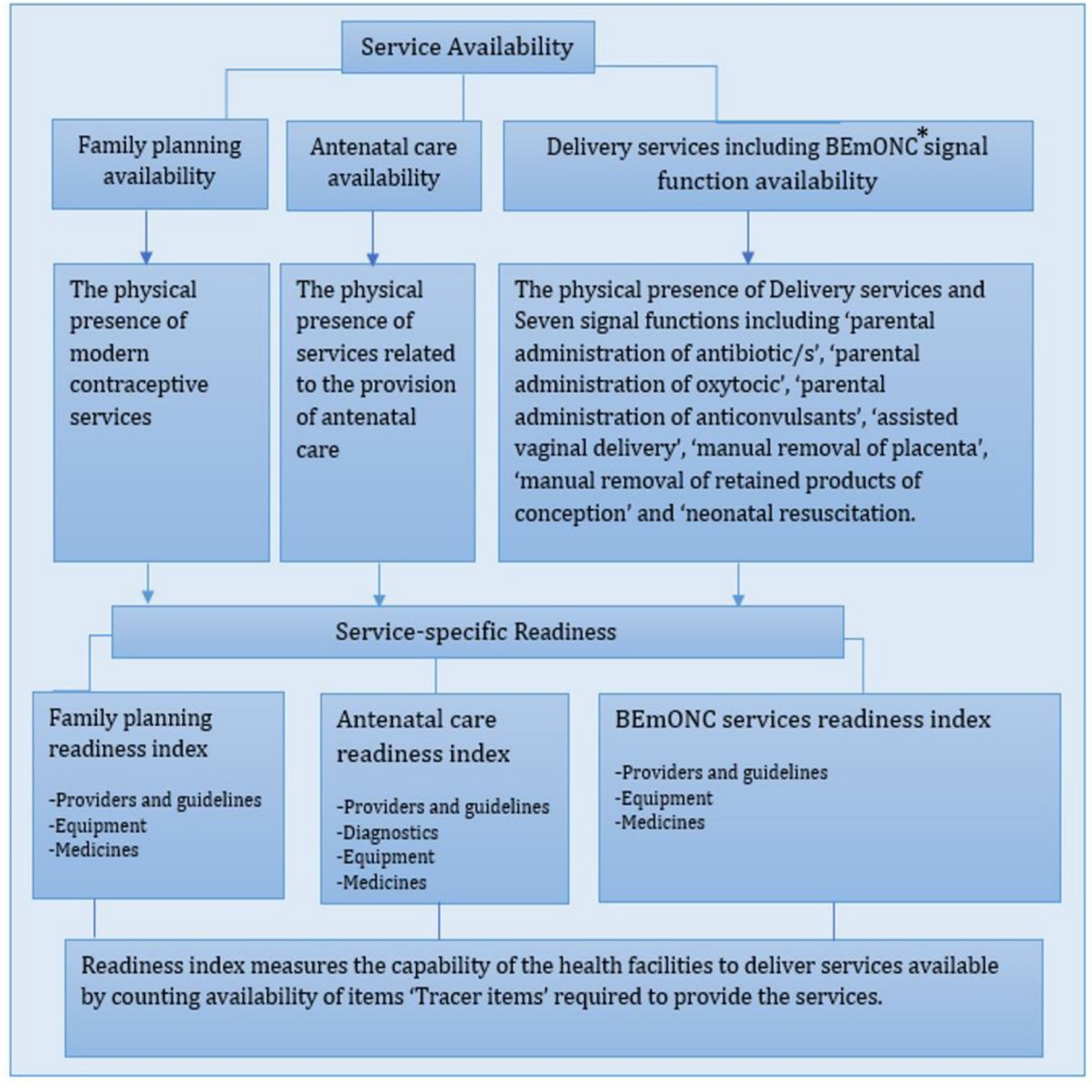

Available organisation characteristics for the health facilities were considered as independent variables. The information related to the number of external supervisory visits in the last three months from the district, regional or national offices (classified as at least one external supervisory visit in the last three months or no/occasional visits). Similarly, the documentation of quality assurance activities, including periodic mortality review, an audit of records or

registers, and/or quality assurance meetings, was coded as present if evidence of documentation was reported and not present if there was no documentation of these activities.

## Data analysis

The data were analysed using SPSS v. 28 (IBM Statistics). The data were weighted with the multiplier provided in the dataset to ensure that the data were proportionally represented as hospitals were oversampled. We used summary statistics to describe the characteristics of health facilities including the availability of specific health services. We calculated mean (standard deviation) for all continuous data and number (%) for the categorical data. We used an independent sample t-test to assess the change in readiness indices from 2015 to 2021.

Multivariable linear regression was used to estimate the effect of various administrative and health service related factors on readiness. The models were fitted to assess the overall effect of independent variables including ecological region, settings (location) of health facilities, managing authority, external supervision, and quality assurance activities, in a combined dataset including data from both years with year of survey included as an independent variable. Furthermore, we also looked at cross-sectional predictors of readiness to compare the impact of each predictor at two different time points. The results are reported as β (95% confidence intervals). We could not assess factors associated with longitudinal change in the indices as the health facilities were de-identified and coded differently in 2015 and 2021. The readiness indices were transformed into percentage values for the regression analysis. The level of significance was set at $p < 0.05$ for all tests.

## Ethical consideration

The 2015 and 2021 NHFS surveys were approved by the Nepal Health Research Council (NHRC) and the ICF Institutional Review Board. The Monash University Human Research Ethics Committee provided ethical approval for this study (HREC number: 33034). We retrieved the data from the DHS program following approval for its use in this study and in accordance with the approved protocol.

## Results

The extracted dataset included 940 facilities from 2015 NHFS and 1565 facilities from 2021 NHFS. Only six facilities (0.6%) had missing information on rural and urban status in 2015. While there were no major differences between facility characteristics in 2015 versus 2021, there was a significantly higher proportion of facilities with regular management meetings in 2015, compared to 2021 (78.3% vs 48.9%, $p < 0.001$) (Table 1).

### Family planning services availability and readiness

Almost all health facilities provided at least one form of family planning service in both years (n = 919/940, 97.8% in 2015 vs n = 1530/1565, 97.8% in 2021) (Table 2). There was 2.9 point increase (95% CI: 1.7 to 4.1, $p < 0.001$) in the Family Planning Readiness Index from 68.0% (SD: 17.0%) in 2015 to 70.9% (SD: 13.5%) in 2021. The staff and guidelines domain for family planning had lowest scores in both years (22.0% (SD: 28.9%) in 2015 vs. 20.7% (SD:30.2%) in 2021). Staff and guidelines and medicines and commodities domain scores were alike between 2015 and 2021.

When we looked at the change in the family planning readiness by provinces, the Gandaki province had lower readiness by -2.2 in 2021 and the Koshi province had the highest achievement by 5.3 (Fig 1).

**Table 1. Characteristics of health facilities in Nepal.**

| Health facilities and characteristics | Subcategories | NHFS 2015 (N = 940) n (%) | NHFS 2021 (N = 1565) n (%) |
|---|---|---|---|
| **Provinces** | Koshi | 163 (17.4) | 262 (16.8) |
| | Madhesh | 171 (18.3) | 246 (15.7) |
| | Bagmati | 187 (20.0) | 321(20.5) |
| | Gandaki | 119 (12.8) | 198 (12.6) |
| | Lumbini | 140 (15.0) | 239 (15.3) |
| | Karnali | 67 (7.1) | 128 (8.2) |
| | Sudurpaschim | 88 (9.5) | 169 (10.8) |
| **Setting*** | Rural | 456 (48.8) | 730 (46.7) |
| | Urban | 479 (51.2) | 834 (53.3) |
| **Ecological region**** | Mountain | 118 (12.6) | 210 (13.4) |
| | Hill | 482 (51.3) | 819 (52.3) |
| | Terai (Low-lying region) | 340 (36.2) | 535 (34.2) |
| **Facility types** | Advanced level public hospital | 22 (2.3) | 27 (1.8) |
| | Peripheral public centres | 849 (90.3) | 1421 (90.8) |
| | Private hospital | 70 (7.4) | 116 (7.4) |
| **Managing authority** | Public | 871 (92.6) | 1448 (92.6) |
| | Private | 70(7.4) | 116 (7.4) |
| **Management meetings** | At least once in three months | 736 (78.3) | 766 (48.9) |
| | None or sometimes | 204 (21.7) | 799 (51.1) |
| **Quality assurance activities** | Documented | 188 (19.9) | 363 (23.3) |
| | None documented | 753(80.1) | 1201 (76.8) |
| **External Supervision** | At least once in the last 4 months | 590 (62.7) | 1036 (66.2) |
| | No supervision | 351 (37.3) | 529 (33.8) |

*Health facilities located in municipality, sub-metropolitan and metropolitan are categorised as urban and health facilities located in village municipalities are categorised as rural.

**Mountain: facilities located in mountainous region of Nepal situated at altitudes of 4,800 metres to 8,839 metres above the sea level, Hill: facilities located in hilly region of Nepal situated at altitudes of 610 metres to 4800 metres above the sea level, Terai: facilities located in the plain region of Nepal situated in the Gangetic plains ranging in altitude of less than 600m above the sea level.

The provision of contraceptives also varied according to the characteristics of health facilities. Specifically, while a significantly higher proportion of public health care facilities provided the oral contraceptive pill compared to private facilities (98% public vs 56% private in 2015 and 98% public vs 52% private in 2021), contrasting results were observed for complete sterilisation (1% public vs 27% private for 2015 and 1% vs 24% for 2021) (S1 Fig).

**Antenatal care availability and readiness.** The majority of health services provided antenatal care at both timepoints (n = 920/940, 97.8% in 2015 and n = 1538/1565, 98.3% in 2021) (Table 2). The Antenatal Readiness Index increased significantly from 49.5% (SD: 17.1%) to 54.1% (SD: 15.9%) between 2015 and 2021 (MD: 4.6, 95% CI:3.263 to 5.937, $p<0.001$). The diagnostics domain had the lowest score of 15.0% (SD:33.8%) in 2015, followed by the staff and guidelines domain of 25.9% (SD: 30.9%). In 2021, the staff and guidelines domain had the lowest score of 19.0% (SD: 27.5%) followed by the diagnostics domain with score of 27.5% (SD: 42.3%). Domain scores for antenatal care were all significantly higher in 2021, except for the staff and guidelines domain which was lower than in 2015. The antenatal readiness increased or remained almost the similar in both years for the provinces. The readiness increased by 8.8 in the Koshi province followed by Sudurpaschim by 6.5 and Madhesh Provinces by 6.5 (Fig 2).

**Table 2. Total and domain readiness indices for family planning, antenatal care and basic emergency obstetric and neonatal care in 2015 and 2021 and change over time.**

| Specific health services | Readiness indices mean score as % (SD) | | Mean difference (95% CI) |
|---|---|---|---|
| | NHFS 2015 N = 940 | NHFS 2021 N = 1565 | |
| **Family Planning** | | | |
| **Availability of services n (%)** | 919 (97.8) | 1530 (97.8) | |
| **Family Planning Readiness Index#** | 68.0 (17.0) | 70.9 (13.5) | 2.9 (1.7 to 4.2*** |
| Staff and guidelines score | 22.0 (28.9) | 20.7 (30.2) | -1.30 (-3.8 to 1.2) |
| Equipment score | 86.6 (34.1) | 96.4 (18.7) | 9.8 (7.8 to 11.9)*** |
| Medicines and commodities score | 95.5 (18.4) | 95.7 (17.1) | 0.2 (-1.2 to 1.6) |
| **Antenatal care** | | | |
| **Availability of services n (%)** | 920 (97.8) | 1538 (98.3) | |
| **Antenatal Readiness Index#** | 49.5 (17.1) | 54.1 (15.9) | 4.6 (3.3 to 5.9)*** |
| Staff and guidelines score | 25.9 (30.9) | 19.0 (27.5) | -6.9 (-9.3 to -4.5)*** |
| Equipment score | 85.9 (34.8) | 96.0 (19.6) | 10.1 (7.9 to 12.3)*** |
| Medicines and commodities score | 71.2 (20.1) | 74.6 (18.9) | 3.4 (1.8 to 4.9)*** |
| Diagnostics score | 15.0 (33.8) | 27.5 (42.3) | 12.5 (9.3 to 15.7)*** |
| **Basic emergency obstetric and neonatal care** | | | |
| **Availability of delivery services n (%)** | 457 (48.6) | 804 (51.4) | |
| **Availability of caesarean section n (%)** | 48 (5.1) | 83 (5.3) | |
| **Basic emergency obstetric and neonatal care Readiness Index#** | 56.7 (16.2) | 58.0 (12.7) | 1.3 (-0.3 to 2.9) |
| Staff and guidelines score | 27.1 (31.3) | 20.0 (27.4) | -7.1 (-10.4 to -3.8)*** |
| Equipment score | 72.2 (15.8) | 81.8 (10.9) | 9.6 (8.1 to 11.1)*** |
| Medicines and commodities score | 70.9 (22.3) | 72.8 (15.5) | 1.9 (-0.2 to 3.9) |

#Scale 0–100% where a higher score indicates greater readiness

*p <0.05

**p <0.01

*** p<0.001 derived from t-test, SD: standard deviation, SE: standard error, CI: Confidence interval

## Delivery services availability and Basic emergency obstetric and newborn care readiness

About half of the health facilities provided delivery services in 2015 and 2021 (n = 457/940, 48.6% in 2015 and n = 804/1565, 51.4% in 2021), although caesarean section was infrequently provided (n = 48 (5.1%) in 2015 and n = 83 (5.3%) in 2021) (Table 2). The Basic Emergency Obstetric and Neonatal Care Readiness Index increased by 1.3 from 56.7% (SD: 16.2%) in 2015 to 58.0% (SD: 12.7%) in 2021 which was statistically insignificant (MD: 1.3, 95% CI -0.316 to 2.916, P = 0.115). The equipment domain score remained similar in both years, medicines and commodities domain scores increased while the staff and guidelines domain score decreased significantly in 2021 from 27.1% in 2015 to 20.0% in 2021. The change in the basic emergency obstetric and newborn care readiness varied across provinces with the observed highest change in Koshi, Karnali and Bagmati whereas the Madhesh had the highest decrease followed by Lumbini province (Fig 3).

Fig 4 displays the proportion of health facilities providing obstetric signal functions in the two surveys. Except for use of uterotonics (85.8% in 2015 vs 88.2% in 2021), provision of all signal functions was lower in 2021 compared to 2015. The availability of basic emergency obstetric and newborn care signal functions also varied across health facilities. Even though peripheral public health facilities comprise 90% of all facilities, a lower proportion of peripheral facilities provided obstetric signal functions compared to private and advanced level public health facilities in 2021 (S2 Fig).

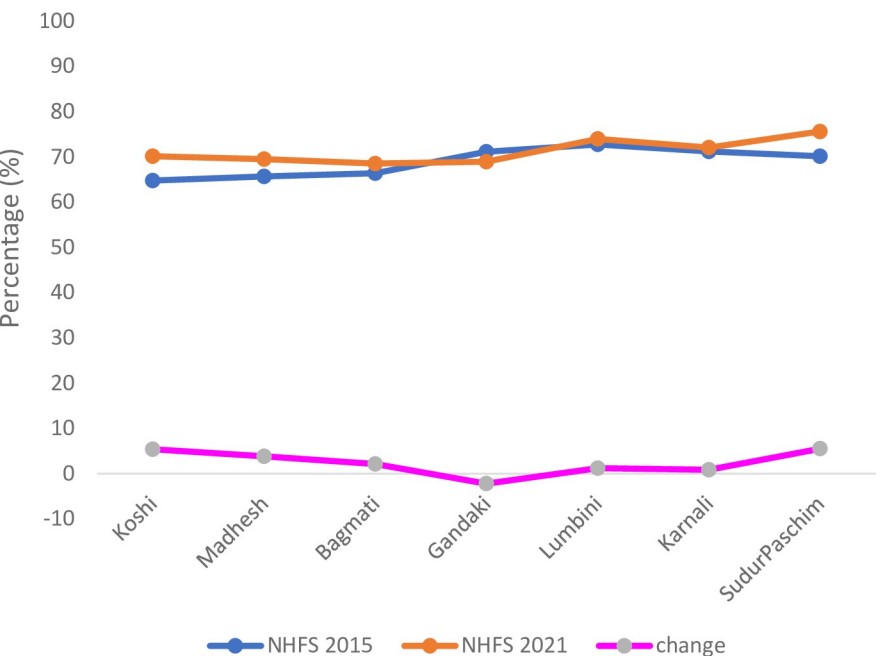

**Fig 1. Family planning readiness index and its change between 2015 and 2021 by provinces.**

## Tracer items for specific health services

There was a sharp decline in the availability of tracer items between 2015 and 2021 including availability of trained family planning providers (from 30.6% to 20.9%), antenatal care guidelines (from 25% to 10.5%), guidelines on childbirth (from 21.8% to 12.8%), and antibiotic eye drops/ointments (from 39.5% to 7.8%) (S2 Table).

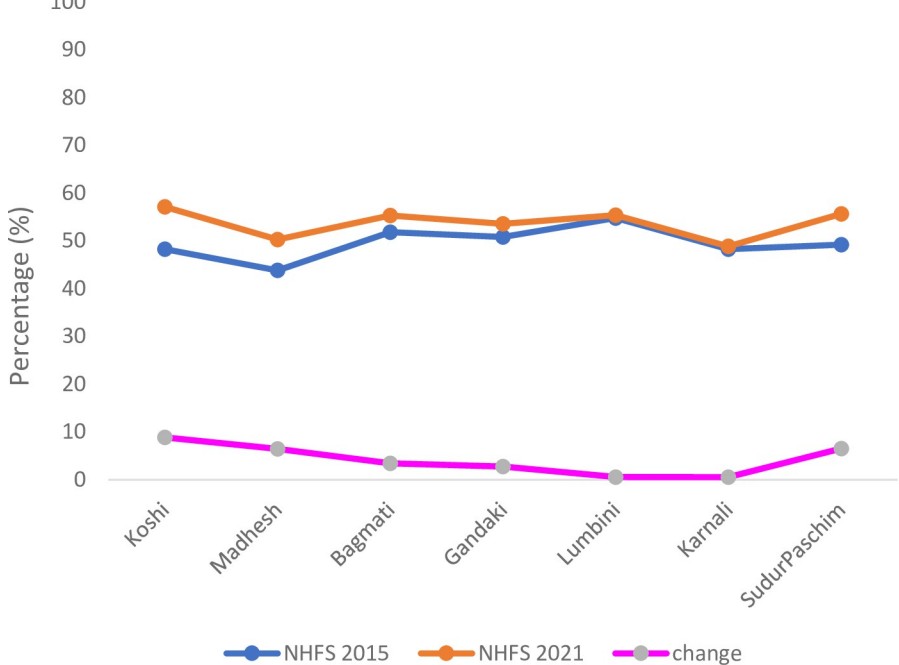

**Fig 2. Antenatal care readiness index and its change between 2015 and 2021 by provinces.**

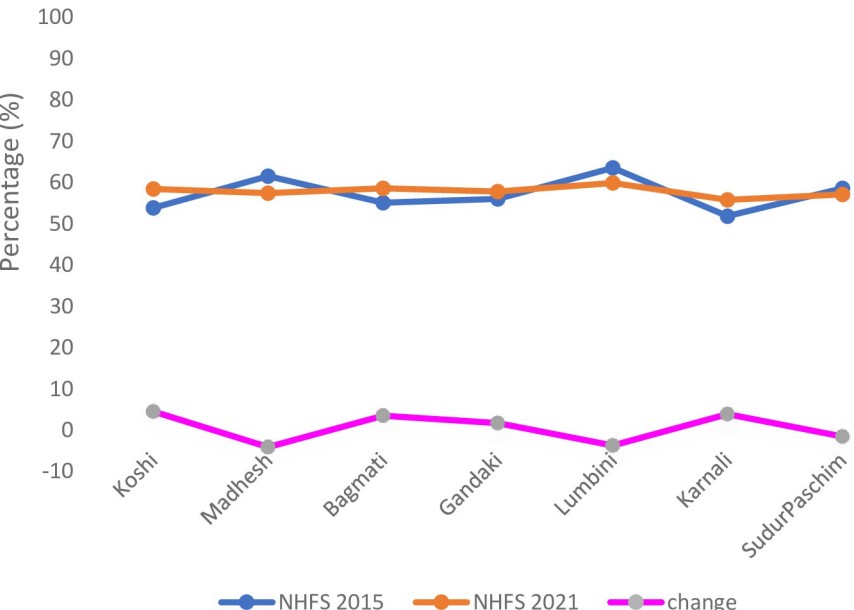

**Fig 3. Basic emergency obstetric and newborn care readiness index and its change between 2015 and 2021 by provinces.**

## Factors affecting the family planning, antenatal care and basic emergency obstetric and newborn care readiness indices

The results of the multivariable linear regression show that antenatal care and family planning indices were higher in 2021 compared to 2015, while there was no year of survey effect on

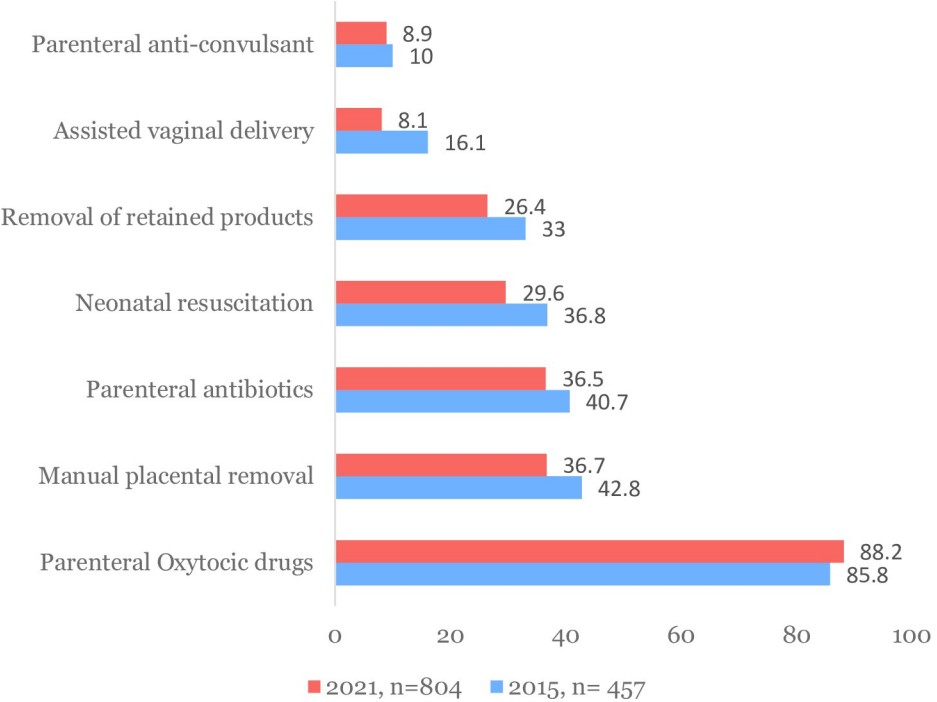

**Fig 4. Provision of obstetric signal functions in the last 3 months (%) in 2015 and 2021.**

**Table 3. Overall effect of geographical and organisational factors on the maternal and neonatal care readiness of Nepalese health facilities from combined datasets of NHFS 2015 and NHFS 2021.**

| Factors | Family Planning readiness | Antenatal care readiness | Basic Emergency Obstetric and Newborn Care readiness |
|---|---|---|---|
| | B (95% CI) | B (95% CI) | B (95% CI) |
| **Ecological region** | | | |
| Terai/Hill | Ref. | Ref. | Ref. |
| Mountain | 0.8 (-1.5 to 3.0) | -1.0 (-3.3 to 1.2) | 0.9 (-1.8 to 3.6) |
| **Setting** | | | |
| Rural | Ref. | Ref. | Ref. |
| Urban | 0.3 (-1.4 to 2.0) | 0.7 (-1.2 to 2.5) | 2.2 (0.1 to 4.3) * |
| **Managing authority** | | | |
| Private | Ref. | Ref. | Ref. |
| Public | 18.6 (15.1 to 22.1) *** | -13.6 (-16.1 to -11.0) *** | 5.7 (2.1 to 9.4) ** |
| **Managing meeting** | | | |
| None or sometimes | Ref. | Ref. | Ref. |
| At least once in 3 months | 2.4 (0.7 to 4.2) ** | 3.9 (2.0 to 5.8) *** | 3.9 (1.7 to 6.2) ** |
| **External supervision** | | | |
| None or sometimes | Ref. | Ref. | Ref. |
| Regular | 0.8 (-0.9 to 2.6) | 3.4 (1.6 to 5.1) *** | 1.5 (-0.8 to 3.8) |
| **Quality assurance activities** | | | |
| None or sometimes | Ref. | Ref. | Ref. |
| Regular | 2.6 (0.8 to 4.5) ** | 4.8 (2.7 to 6.8) *** | 3.5 (1.1 to 5.9) ** |
| **Year** | | | |
| 2015 | Ref. | Ref. | Ref. |
| 2021 | 0.5 (0.2 to 0.8) ** | 0.9 (0.6 to 1.2) *** | 0.3. (-0.1 to 0.7) |

Significant at

*p-value <0.05

** p<0.01

*** p<0.001

basic emergency obstetric and newborn care. Facilities with regular management meetings, and quality assurance activities had higher family planning, antenatal care and basic emergency obstetric and newborn care readiness (Table 3). Publicly managed facilities had significantly lower antenatal readiness (B: -13.6, 95% CI -16.1 to -11.0, p<0.001) but higher family planning readiness (B: 18.6, 95% CI 15.1 to 22.1, p<0.001) and basic emergency obstetric and newborn care readiness (B: 5.7, 95% CI 2.1 to 9.4, p = 0.002) compared to private facilities. Facility location (either based on ecological region or rurality setting) did not affect the readiness score for all indices, with the exception of basic obstetric and newborn care where facilities located in the urban area had a higher readiness score (B: 2.2, 95% CI 0.1 to 4.3, p: 0.043). External supervision had a positive effect only on antenatal care readiness (B: 3.4, 95% CI 1.6 to 5.1, p< 0.001) (Table 3).

The results of the stratified analysis show that in 2015, practices located in urban areas had higher family planning readiness (B: 3.5, 95% CI 0.4 to 6.7, p: 0.029) and antenatal readiness (B: 5.4, 95% CI 2.6 to 8.2, p<0.001) as well as higher basic emergency readiness index (B: 5.7, 95% CI 1.8 to 9.5, p<0.001) (Table 4). The results based on the 2021 survey show that compared with practices located in rural areas, practices located in urban areas had lower family planning readiness (B: -1.6, 95% CI -2.9 to -0.3, p = 0.018) and antenatal care readiness (B: -2.1, 95% CI -3.7 to 6.7, p = 0.029). Managing authority and rurality setting did not have a significant effect on basic emergency obstetric and newborn care (Table 4).

**Table 4. Factors associated with maternal and neonatal care readiness, stratified by year.**

| Factors | B (95% CI) | B (95% CI) |
|---|---|---|
| **Family planning readiness index** | **NHFS 2015** | **NHFS 2021** |
| **Ecological region** | | |
| Terai/Hill | Ref. | Ref. |
| Mountain | 0.7(-2.5 to 3.9) | 1.1(-0.8 to 2.9) |
| **Setting** | | |
| Rural | Ref. | Ref. |
| Urban | 3.5(1.3 to 5.7)** | -1.6(-2.9 to -0.3)* |
| **Managing authority** | | |
| Private | Ref. | Ref. |
| Public | 19.2(14.4 to 24.0)*** | 18.5(15.7 to 21.4)*** |
| **Managing meeting** | | |
| None or sometimes | Ref. | Ref. |
| At least one in 3 months | 4.1(1.5 to 6.7)** | 1.8(0.5 to 3.1)** |
| **External supervision** | | |
| None or sometimes | Ref. | Ref. |
| Regular | -0.02(-2.2 to 2.2) | 1.3(-0.07 to 2.6) |
| **Quality assurance activities** | | |
| None or sometimes | Ref. | Ref. |
| Regular | 2.7(0.1 to 5.4)* | 2.6(1.1 to 4.1)*** |
| **Antenatal care readiness index** | NHFS 2015 (n = 451) | NHFS 2021 (n = 537) |
| **Ecological region** | | |
| Terai/Hill | Ref. | Ref. |
| Mountain | 0.3(-2.8 to 3.5) | -1.8(-4.1 to 0.5) |
| **Setting** | | |
| Rural | Ref. | Ref. |
| Urban | 5.4(3.2 to 7.6)*** | -2.1(-3.7 to -0.5)** |
| **Managing authority** | | |
| Private | Ref. | Ref. |
| Public | -15.3(-19.6 to -11.0)*** | -12.5(-15.6 to -9.4)*** |
| **Managing meeting** | | |
| None or sometimes | Ref. | Ref. |
| At least one in 3 months | 3.4(0.8 to 5.9)** | 4.1(2.6 to 5.7)*** |
| **External supervision** | | |
| None or sometimes | Ref. | Ref. |
| Regular | 3.9(1.8 to 6.1)*** | 2.9(1.3 to 4.6)*** |
| **Quality assurance activities** | | |
| None or sometimes | Ref. | Ref. |
| Regular | 4.7(2.1 to 7.2) *** | 4.7(2.8 to 6.5)*** |
| **BEmONC readiness index** | NHFS 2015 (n = 451) | NHFS 2021 (n = 803) |
| **Ecological region** | | |
| Terai/Hill | Ref. | Ref. |
| Mountain | -1.4(-5.6 to 2.8) | 2.2(-0.9 to 4.5) |
| **Setting** | | |
| Rural | Ref. | Ref. |
| Urban | 5.7 (2.5 to 8.8)*** | 0.1(-1.7 to 1.9) |
| **Managing authority** | | |
| Private | Ref. | Ref. |

(*Continued*)

**Table 4.** (Continued)

| Factors | B (95% CI) | B (95% CI) |
|---|---|---|
| Public | 10.3(5.1 to 15.5)*** | 2.6(-0.8 to 6.0) |
| **Managing meetings** | | |
| None or sometimes | Ref. | Ref. |
| At least one in 3 months | 2.7(-1.2 to 6.6) | 4.6(2.8 to 6.4)*** |
| **External supervision** | | |
| None or sometimes | Ref. | Ref. |
| Regular | 1.4(-1.9 to 4.7) | 1.6(-0.27 to 3.4) |
| **Quality assurance activities** | | |
| None or sometimes | Ref. | Ref. |
| Regular | 3.6(-0.4 to 7.3) | 3.5(1.6 to 5.3)*** |

Significant at *p-value <0.05

** p<0.01

*** p<0.001

## Discussion

This study reports the readiness and availability of key maternal and newborn health services in Nepal (specifically, family planning, antenatal care, and basic emergency obstetric and newborn care services) and changes in readiness indices from 2015 to 2021. We report province-specific readiness and the contribution of geographic and organisational factors to readiness, to inform health service planning and identify opportunities where more resources are required. The NHFS data showed that almost all health facilities offered family planning and antenatal care services in both 2015 and 2021; in contrast, only half of the facilities provided delivery services in 2021 with little change since 2015. Concerningly, even when health facilities offered these services, they were not fully prepared, as reflected in low readiness indices. Of further concern is that only small improvements in readiness indices were observed for these critical maternal and newborn services (2.9% for family planning, 4.6% for antenatal care, and 1.3% for basic emergency obstetric and newborn care). These observed improvements are unlikely to represent meaningful changes in the readiness and quality of care. The consistently low scores observed for the staff and guidelines domain (for family planning, antenatal care, and basic emergency obstetric and newborn care) and the diagnostics domain (for antenatal care) highlight important opportunities for system-wide improvements.

Greater readiness of health facilities has been linked with decreased mortality in diseases such as malaria [35] and neonatal mortality [36], and a lack of readiness is identified as an impediment to the delivery of quality care [37]. This is reinforced by research from Nepal, which found that women receive higher quality antenatal care services when they attend health facilities that are better equipped [38]. Impaired organisational capacities (which can include insufficient supplies and equipment, along with suboptimal knowledge and skills) have also been identified as key barriers to the timely delivery of maternal health services in countries such as Paraguay and Ghana [39, 40]. Similarly, a meta-review (including 92 systematic reviews) of facilitators and barriers to quality care in maternal, newborn and child health demonstrated that inadequate health facility resources and variable standards of clinical guidelines were barriers to quality care in maternal and newborn and child health [41].

Our study suggests that significant improvements in maternal and neonatal health outcomes in Nepal are unlikely to be realised given only small improvements in the availability of delivery services and readiness indices for maternal health care and a decrease in the delivery

of signal functions from 2015 to 2021 [23, 42]. The marginal improvement in readiness for basic emergency obstetric and newborn care (from 56.7% in 2015 to 58.0% in 2021) is of particular concern and is in keeping with other reports have also highlighted the relatively low readiness for this important service in both 2015 [25] and 2021 [29]. A small study conducted in the Jumla district of Nepal which found low readiness for maternal and newborn care, reported that 16% of health facilities were unable to provide treatment due to lack of medicines, while 80% needed to refer patients to higher-level health facilities for required care [43]. It has also been estimated that only 8% of women expected to have major obstetric complications were able to be treated at emergency care obstetric centres in 2020/2021 [21].

Our study showed regional variation in readiness scores, with some provinces showing a small deterioration in readiness over time. Compared to 2015, the Madhesh, Lumbini and Sudurpaschim provinces had reduced readiness for providing obstetric and newborn care and the Gandaki province had reduced readiness for providing family planning services in 2021. These data provide tangible guidance for local governments, by highlighting provinces where greater resources are required to improve health facility readiness. We also found that public facilities had a greater readiness for providing family planning and delivery services in 2015 and 2021 and that private facilities had a greater readiness for antenatal care at both time-points. This could be due to a greater availability of diagnostic facilities in private health facilities, as these services contribute to the readiness index.

A recent study reported that private facilities in Nepal had substantially lower readiness for providing basic emergency obstetric and neonatal care in 2021, but did not examine other readiness indices [29]. As basic health services and reproductive health are granted by the constitution of Nepal [44], private health facilities should focus on increasing readiness for all types of basic reproductive and maternity services at every level. We also observed that health facilities in rural areas had lower readiness for providing basic emergency obstetric and newborn care, compared with facilities in urban areas. This is in contrast to recent findings which reported lower readiness among urban centres [29]. Inequity between regions, urban and rural settings, and public and private settings should be taken into consideration by local administrative authorities as part of health service planning.

In line with the Sustainable Development Goals, the Nepal safe motherhood and newborn health road map 2030 aims to decrease maternal and newborn mortality by providing high-quality maternal and newborn health services [45]. However, our analysis of 2015 and 2021 NHFS data indicates that urgent investment in Nepalese health facilities is needed (for example, enhanced staff training, making national guidelines available, and investing in diagnostic facilities where these are lacking) to improve readiness for the delivery of quality maternal health services. The introduction of administrative activities (such as quality assurance activities and regular management meetings, which were associated with significantly higher readiness in this study) may also offer opportunities for improving readiness. It has previously been suggested that increasing service hours, introducing the periodic evaluation of maternal and newborn facilities, and concurrent investment in raising awareness in the community for utilisation of the available health services is also needed [29].

It is also helpful to consider the study findings in the context of data from other developing countries which face similar health system and resourcing challenges to Nepal. Compared to sub-Saharan African countries, a higher proportion of health facilities in Nepal offered family planning and antenatal care services [46, 47]. Readiness for antenatal care readiness in Nepal in 2021 (54.1%) is comparable to the readiness for antenatal care reported for neighbouring Bangladesh (54.4%) in 2017 [48]. Readiness for basic emergency obstetric and newborn care in Nepal is lower (58.0% in 2021) compared to Ethiopia (62.7%) [37] but higher when compared to readiness reported for Tanzania (40.1%) [49]. Benchmarking availability and readiness data

for Nepal against other developing countries and identifying successful strategies that have led to greater health facility readiness elsewhere may also be useful.

## Strengths, limitations and implications for future research

This analysis is based on nationally representative data collected using standardised questionnaires. As such, the findings can be generalised more broadly to the overall Nepalese health system and health facilities. We examined health facility readiness in 2015 and 2021 for three key indices pertinent to maternal and newborn care, and examined the contribution of geographical and organisational factors to readiness. This study is an advance over recent reports that describe 2021 NHFS data or provide limited information on service readiness, availability, and signal functions [29, 30]. As the 2015 and 2021 surveys were cross-sectional in design, we were not able to determine pairwise factors that influenced changes over time. Further research could explore longitudinal changes at the individual facility level. Understanding the proportion of women and newborns referred to other health facilities due to the unavailability of specific tracer items is also an area of further research, as is qualitative research to understand the barriers and enablers to addressing facility readiness from the perspective of healthcare providers. Furthermore, some indicators such as the density of health facilities and healthcare personnel could not be assessed due to insufficient data. This makes it difficult to determine whether there are sufficient healthcare personnel in health facilities to cater to the needs of the target population.

## Conclusion

While most health facilities in Nepal provided family planning services and antenatal care, delivery services were available in only half of the facilities, with no change in availability observed over the study period. Little improvement in readiness for family planning and basic emergency obstetric and newborn care was observed from 2015 to 2021, although a small improvement in readiness for antenatal care was evident. Readiness for antenatal care remained the lowest in 2021, followed by readiness for basic emergency obstetric and newborn care. Greater investment in training staff, the introduction of quality assurance activities, regular management meetings, and periodic internal and external evaluation of health services may assist in increasing readiness. Increased provision of diagnostic facilities (where these are lacking) and greater attention to addressing sustained inequities in readiness and service availability between provinces and between public and private facilities may also assist in increasing readiness.

## Supporting information

**S1 Checklist. STROBE 2007 (v4) statement—checklist of items for cross-sectional studies.**
(DOCX)

**S1 Fig. Types of contraceptives provided according to background characteristics of the health facilities in 2015 and 2021.**
(DOCX)

**S2 Fig. Availability of basic emergency obstetric and newborn care signal functions in 2015 and 2021 according to background characteristics of the health facilities.**
(DOCX)

**S1 Table. Tracer items used in the calculation of service specific readiness indexes.**
(DOCX)

**S2 Table. Presence of tracer items for family planning, antenatal care and basic emergency obstetric and newborn care services 2015 vs 2021.**
(DOCX)

## Author Contributions

**Conceptualization:** Pramila Rai, Ilana N. Ackerman, Denise A. O'Connor, Rachelle Buchbinder.

**Data curation:** Pramila Rai.

**Formal analysis:** Pramila Rai, Alexandra Gorelik.

**Methodology:** Pramila Rai.

**Supervision:** Ilana N. Ackerman, Denise A. O'Connor, Rachelle Buchbinder.

**Validation:** Ilana N. Ackerman, Denise A. O'Connor, Alexandra Gorelik, Rachelle Buchbinder.

**Writing – original draft:** Pramila Rai.

**Writing – review & editing:** Pramila Rai, Ilana N. Ackerman, Denise A. O'Connor, Alexandra Gorelik, Rachelle Buchbinder.

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
