## [Decision Letter · Decision Letter 0]

12 Jun 2023

PONE-D-23-12847Health facility availability and readiness for family planning and maternity and neonatal care services in Nepal: Analysis of cross-sectional survey dataPLOS ONE

Dear Dr. Rai,

Thank you for submitting your manuscript to PLOS ONE. After careful consideration, we feel that it has merit but does not fully meet PLOS ONE’s publication criteria as it currently stands. Therefore, we invite you to submit a revised version of the manuscript that addresses the points raised during the review process. Please see my comments in comment section attached herewith. Requesting you to revise the paper based on reviewer's and my comments.

We look forward to receiving your revised manuscript.

Kind regards,

Kanchan Thapa, MPH, MPhil

Academic Editor

PLOS ONE

“The author(s) received no specific funding for this work. Pramila Rai is supported with Monash International Postgraduate Research Scholarship and Monash Graduate Scholarship for her PhD.”

Additional Editor Comments:

Dear Authors,

I really enjoyed reading your paper. However, there are something which can improve your paper further. Please take care of all the comments from reviewers. Also, take care of these comments from my side.

In Abstract section, analytical perspectives of logistic regression is lacking. You have only used descriptive data to show the change over the time. There are limited information from logistic regression in the section.

Can you please review the Method section and especially in the data source section. Also in ethical section, there is special provision of DHS program to use the data set for analysis, have you followed the procedure for data use. Please also include the statement, how you downloaded data set from DHS program and when.

Results:

Table 1.

There can be better way of writing the table characteristics such as 2015 into NFHS 2015 or something better. Also, take care of N and n difference while reporting the research results.

Table 2. Same as above, but you have used n here. What is the difference between these two. What is MD 95(CI) mean?

Table 3. based on which data the results are reported? Also advised to see other results how to report the findings from logistic regression and reference.

Table 4. Same as previous comments

Also requesting you to review the whole paper once from all the authors.

Reviewers' comments:

Reviewer's Responses to Questions

**Comments to the Author**

1. Is the manuscript technically sound, and do the data support the conclusions?

Reviewer #1: Yes

Reviewer #2: Partly

Reviewer #3: Yes

2. Has the statistical analysis been performed appropriately and rigorously? 

Reviewer #1: Yes

Reviewer #2: Yes

Reviewer #3: No

3. Have the authors made all data underlying the findings in their manuscript fully available?

Reviewer #1: Yes

Reviewer #2: Yes

Reviewer #3: Yes

4. Is the manuscript presented in an intelligible fashion and written in standard English?

Reviewer #1: Yes

Reviewer #2: Yes

Reviewer #3: Yes

5. Review Comments to the Author

Reviewer #1: The authors assessed Health facility availability and readiness for family planning and maternity and

neonatal care services in Nepal By relying on Nepal Health Facility Survey (NHFS)

data collected in 2015 and 2021. The authors further relied on WHO Service Availability and Readiness Assessment (SARA) Methods to determine the key variables necessary to attain the specific objectives of the study.

I took my time to go through the WHO SARA document while comparing same with what the authors did.

Per the WHO's SARA document, Availability is defined as the Physical presence of services while Readiness is referred to as the capacity of the facility to deliver those services.

Readiness and Availability were the two important dimensions in the study because they were the indicators the authors used to check family planning and maternity and neonatal care services being rendered in Nepal.

The WHO SARA document has highlighted that to generate reliable information on Service Availability then facility density, health worker density and service utilization should be assessed.

Also to generate reliable information on Service Readiness then Basic Amenities, Equipment & Supplies, Diagnostics, Essential medicines and commodities should all be assessed. Since such assessments will be the measurement scale to tell whether or not the there are Service Availability and Service Readiness.

Afterwards then the Specific Service Readiness Areas such as: Family Planning, Antenatal care, Obstetric care, Neonatal care and child health (curative and immunization), HIV, PMTCT, TB, Malaria, Chronic Diseases can be checked as to whether they are being rendered adequately or not.

Authors highlighted the facility types in table 1. However, the facility density was not captured. It is recommended that authors fuse in this data to enrich their findings. This is due to the fact that the facility density will enable us to see whether or not the listed facility types possess the requisite space and capability to render the key services of interest to the authors.

Also, health worker density across the listed facilities were not highlighted by the authors. It is again recommended that the authors include this important data in their results presentation. As the health worker density will enable us to know whether or not the listed facilities have the requisite numbers of health professionals to render the key services of interest.

Except the two important observations made above, authors rigorously complied with the methodology proferred in the WHO SARA guidelines.

Authors should modify their work to take cognizance of the recommendations made above.

In a nutshell, it is a very good work done by the authors.

Reviewer #2: I am very pleased to have the opportunity of reviewing the manuscript submitted to PLOS ONE Journal.

Thanks to the authors for carrying out this good study. The manuscript has enough potential to publish as a scientific work.

Best Regards,

Reviewer #3: 1. The manuscript was technically sound enough since the right methodology was used. WHO SARA tool was appropraite for the study

2. Analysis for objective one was good. However, the analysis for the secondary objective was unclear. You need to define your dependent and independent variables used fo the regression model clearly.

3. Findings supported study objectives

6. PLOS authors have the option to publish the peer review history of their article (what does this mean?). If published, this will include your full peer review and any attached files.

Reviewer #1: **Yes: **ANTWI JOSEPH BARIMAH

Reviewer #2: No

Reviewer #3: No

---

## [Author Response · Author response to Decision Letter 0]

3 Jul 2023

Response to reviewers’ comments

We thank the Editor and Peer Reviewers for their helpful comments on our manuscript. Our point-by-point response to each comment, along with page and line numbers as they appear in the clean revised version, are below. We look forward to your decision about publication of the manuscript.

Kind regards

Pramila Rai, on behalf of the author team

Academic Editor 

Journal requirements

1. “Please ensure that your manuscript meets PLOS ONE's style requirements, including those for file naming.”

Author response: This has been addressed as requested. 

“The author(s) received no specific funding for this work. Pramila Rai is supported by a Monash International Postgraduate PhD Research Scholarship and Monash Graduate Scholarship.

Author response: We have updated our financial disclosure statement (in the cover letter) as requested. This now states:

“The authors received no specific funding for this work. Ms Rai is supported by a Monash University International Postgraduate Research Scholarship and Graduate Scholarship for her doctoral research. Dr Buchbinder is supported by an Australian NHMRC Investigator Fellowship (APP1194483). The funders had no role in study design, data collection and analysis, decision to publish, or preparation of the manuscript.”

3. We note that you have indicated that data from this study are available upon request. PLOS only allows data to be available upon request if there are legal or ethical restrictions on sharing data publicly. For more information on unacceptable data access restrictions, please see http://journals.plos.org/plosone/s/data-availability#loc-unacceptable-data-access-restrictions. In your revised cover letter, please address the following prompts:

Author response: We have updated our financial disclosure statement (in cover letter) as requested. This now states:

“The data used in this study are owned by the Demographic and Health Surveys (DHS) Program and re-distribution of DHS data is not permitted by the DHS program. These data are available upon request and approval from the DHS Program. Data requests can be made through the link: https://dhsprogram.com/data/new-user-registration.cfm.”

Author response: Separate captions for each figure have been added.

Author response: This has been amended as requested. Matching in-text citations now appear in lines 164 279, 322 and 331. 

Author response: The reference list has been checked as requested to ensure it is complete and correct. We confirm it does not contain any retracted papers. 

7. In Abstract section, analytical perspectives of logistic regression are lacking. You have only used descriptive data to show the changeover the time. There is limited information from logistic regression in the section.

Author response: This has been addressed as requested. We have updated the sentences (pages 2 to 3, lines 38 to 42 of the abstract) outlining the effect as observed in the regression analysis. We have also kept the paper word limit in mind. 

8. Can you please review the Method section and especially in the data source section? Also, in ethical section, there is special provision of DHS program to use the data set for analysis, have you followed the procedure for data use. Please also include the statement, how you downloaded data set from DHS program and when.

Author response: This has been addressed as requested. We have updated the Methods/ Data Source section (page 6, lines 107 to 109 as follows):

“The NHFS 2015 and 2021 data for this project were retrieved between May-September 2022 and followed the standard procedures for data use. Access to the data was granted to the first author (PR) by the DHS program for this specific project.”

The Methods/ Ethical considerations section (page 11, lines 224 to 230) has been updated as follows:

“The 2015 and 2021 NHFS surveys were approved by the Nepal Health Research Council (NHRC) and the ICF Institutional Review Board. The Monash University Human Research Ethics Committee provided ethical approval for this study (HREC number: 33034). We retrieved the data from the DHS program following approval for its use in this study and in accordance with the approved protocol. ”

9. Results: Table 1. There can be better way of writing the table characteristics such as 2015 into NFHS 2015 or something better. Also, take care of N and n difference while reporting the research results.

Author response: This has been addressed as requested. Labels for columns 3 and 4 in Table 1 (page 12) have been updated for clarity. N (total eligible sample) and n (those with specific characteristics within the total eligible sample) are consistently reported throughout the manuscript. 

10. Table 2. Same as above, but you have used n here. What is the difference between these two. What is MD 95(CI) mean? 

Author response: This has been addressed as requested. The labels for NHFS 2015 and NHFS 2021 in columns 2 to 3 of Table 2 (page 14) and N have been updated. MD (95% CI) refers to the mean difference in domain scores between the two surveys (NHFS 2015 and NHFS 2021) and their 95% confidence interval. The label for “MD (95% CI)” in the last column in Table 2 has been updated to “Mean difference (95% CI)” for clarity. The full form of SE, SD, and CI are provided in the footnotes of the table. 

11. Table 3. based on which data the results are reported? Also advised to see other results how to report the findings from logistic regression and reference.

Author response: This has been clarified as requested. The results are based on the combined datasets from NHFS 2015 and 2021. The title for Table 3 has been updated to reflect this (page 18. Given we used multivariable linear regression, we have used the beta coefficient to report the findings in the text. We have reviewed the PLOS ONE article by Acharya et. al [1] and have followed a similar approach to report our findings (except we use confidence intervals instead of standard errors as a measure of variation). 

12. Table 4. Same as previous comments

Author response: See response to comment #11 above. Table 4 has been formatted to be consistent with Table 3. 

13. Also requesting you to review the whole paper once from all the authors.

Author response: This has been done. All authors have reviewed and approved the revised manuscript. 

 

Reviewer 1

The authors assessed Health facility availability and readiness for family planning and maternity and neonatal care services in Nepal by relying on Nepal Health Facility Survey (NHFS) data collected in 2015 and 2021. The authors further relied on WHO Service Availability and Readiness Assessment (SARA) Methods to determine the key variables necessary to attain the specific objectives of the study.

I took my time to go through the WHO SARA document while comparing same with what the authors did. Per the WHO's SARA document, Availability is defined as the Physical presence of services while Readiness is referred to as the capacity of the facility to deliver those services. Readiness and Availability were the two important dimensions in the study because they were the indicators the authors used to check family planning and maternity and neonatal care services being rendered in Nepal.

The WHO SARA document has highlighted that to generate reliable information on Service Availability then facility density, health worker density and service utilization should be assessed. Also to generate reliable information on Service Readiness then Basic Amenities, Equipment & Supplies, Diagnostics, Essential medicines and commodities should all be assessed. Since such assessments will be the measurement scale to tell whether or not the there are Service Availability and Service Readiness. Afterwards then the Specific Service Readiness Areas such as: Family Planning, Antenatal care, Obstetric care, Neonatal care and child health (curative and immunization), HIV, PMTCT, TB, Malaria, Chronic Diseases can be checked as to whether they are being rendered adequately or not.

Authors highlighted the facility types in table 1. However, the facility density was not captured. It is recommended that authors fuse in this data to enrich their findings. This is due to the fact that the facility density will enable us to see whether or not the listed facility types possess the requisite space and capability to render the key services of interest to the authors.

Also, health worker density across the listed facilities were not highlighted by the authors. It is again recommended that the authors include this important data in their results presentation. As the health worker density will enable us to know whether or not the listed facilities have the requisite numbers of health professionals to render the key services of interest.

Except the two important observations made above, authors rigorously complied with the methodology proferred in the WHO SARA guidelines.

Authors should modify their work to take cognizance of the recommendations made above.

In a nutshell, it is a very good work done by the authors.

Author response: Thank you for your comments. While we agree that calculating the density of health facilities and health workers could have been useful, we were unable to obtain adequate data for these indicators. Nonetheless, availability of trained and supervised staff was one of the tracer items for calculating the readiness indices and we have included the scores for staff availability for each specific service (i.e. family planning, antenatal care, basic emergency obstetric and newborn care) in the supplementary file (S2 Table). In addition, we have added a statement about this as a limitation of the study (page 24-25, lines 470 to 473).

Reviewer 2

1. I am very pleased to have the opportunity of reviewing the manuscript submitted to PLOS ONE Journal. Thanks to the authors for carrying out this good study. The manuscript has enough potential to publish as a scientific work. Best Regards.

Author response: Thank you for your review and comments. 

Reviewer 3

1. The manuscript was technically sound enough since the right methodology was used. WHO SARA tool was appropriate for the study. Analysis for objective one was good. However, the analysis for the secondary objective was unclear. You need to define your dependent and independent variables used for the regression model clearly. Findings supported study objectives

Author response: Our secondary objective was to assess the factors associated with service-specific readiness of health facilities (page 6, line 102) for which we employed multivariable linear regression. The dependent variables were the family planning readiness index, antenatal care readiness index and basic emergency obstetric and newborn care readiness index. These are described in detail on pages 8-9, lines 146 to 180, in the manuscript. Detailed information on the independent variables fitted in the multivariable linear regression, including ecological regions, provinces, settings (rural/urban), type of managing authority, external supervision, and quality assurance activities are provided on pages 9-10, lines 184 to 201 We have also clarified the independent variables fitted in the multivariable linear regression in the Data Analysis section on page 11, lines 213-218. 

References: 

1. Acharya K, Subedi D, Acharya P: Health facility readiness to provide integrated Family Planning, Maternal and Child Health (FPMCH) services in Nepal: Evidence from the comprehensive health facility survey. PLoS One 2022, 17(2):e0264417.

---

## [Editor Report · Decision Letter 1]

19 Jul 2023

Health facility availability and readiness for family planning and maternity and neonatal care services in Nepal: Analysis of cross-sectional survey data

PONE-D-23-12847R1

Dear Dr. Rai,

We’re pleased to inform you that your manuscript has been judged scientifically suitable for publication and will be formally accepted for publication once it meets all outstanding technical requirements.

Kind regards,

Kanchan Thapa, MPH, MPhil

Academic Editor

PLOS ONE

Additional Editor Comments (optional):

Dear Authors,

Thank you for revised version of the paper. I would like to thank you for your hard works. At the same time, requesting you to take care of use of N and n throughout the table. You have revised in the other section and included your sample size as n but still there are some N in tables.

Also, please format the table as per the guidelines of PLOS One.
---

## [Editor Report · Acceptance letter]

28 Jul 2023

PONE-D-23-12847R1 

Health facility availability and readiness for family planning and maternity and neonatal care services in Nepal: Analysis of cross-sectional survey data 

Dear Dr. Rai:

I'm pleased to inform you that your manuscript has been deemed suitable for publication in PLOS ONE. Congratulations! Your manuscript is now with our production department. 

Kind regards, 

on behalf of

Mr. Kanchan Thapa 

Academic Editor

PLOS ONE